# Process evaluation of the HIV+D intervention for integrating the management of depression in routine HIV care in Uganda

**Joshua Ssebunnya**[1]*, **Rutakumwa Rwamahe**[1], **Richard Mpango**[1],
**Leticia Kyohangirwe**[1,2], **Christine Tusiime**[1,2], **Hafsa Sentongo**[3], **Pontiano Kaleebu**[1],
**Vikram Patel**[4], **Eugene Kinyanda**[1]

**1** Medical Research Council/Uganda Virus Research Institute & London School of Hygiene and Tropical Medicine Uganda Research Unit, Mental Health Research Unit, Entebbe, Uganda, **2** Butabika National Referral Mental Hospital, Kampala, Uganda, **3** Mental Health Division, Ministry of Health, Kampala, Uganda, **4** Department of Global Health and Social Medicine, Harvard Medical School, Boston, Massachusetts, United States of America

* joy95h@yahoo.co.uk

**Data Availability Statement:** De-identified data from which this manuscript has been produced will be made available upon request, but the full data

## Abstract

HIV/AIDS continues to be one of the leading global health challenges, having claimed over 40 million lives so far. People infected with HIV are more likely to develop depression, leading to several negative behavioural and clinical outcomes. Studies have reported exceptionally high prevalence of depressive symptoms among people living with HIV/AIDS (PLWHA), making a case for integrating mental healthcare in routine HIV care. The HIV+D program was implemented in Uganda as an intervention model for integrating the management of depression in HIV care. Process evaluation was carried out to understand the contextual factors and explain the outcomes of the intervention. This was a qualitative study, conducted in 28 out of the 40 public health facilities in 3 districts, where the intervention was undertaken. A total of 82 participants, including the implementers and beneficiaries of the intervention were purposively selected for key informant interviews and focus group discussions. Conventional content analysis was done focusing on 6 aspects including reach, effectiveness, acceptability, implementation fidelity, maintenance (sustainability) and contextual factors that affected the intervention. The intervention was well received and believed to address a real need of the intended beneficiaries; many of whom described it as a program for helping the depressed PLWHA to deal with their depression. The implementers were said to have adhered to the intervention protocol, without major program adaptations, suggesting strong implementation fidelity. The respondents cited several positive health outcomes that resulted from the intervention, for both the beneficiaries and the implementers. Several contextual factors affected the implementation. Although it met the beneficiaries' approval, and was considered relevant and acceptable, they expressed uncertainty over sustainability of the intervention in a health system characterized by understaffing, resource constraints and several other challenges. The intervention was generally implemented as intended, resulting in several positive health outcomes.

set may not be shared due to the qualitative and potentially identifiable nature of the raw data (eg, transcripts). Request for data access should be made to the UVRI – REC Chairperson: Mr. Tom Lutalo, tomlutalo@gmail.com and committee member, Mr. Wilber Ssembajjwe, Wilber. ssembajjwe@mrcuganda.org.

**Funding:** The study was supported by an unrestricted grant from the Wellcome Trust through a Senior Research Fellowship in Public Health and Tropical Medicine to EK, reference number 205069/Z/16/Z. The funders had no influence on the design or conduct of the study and were not involved in data collection or analysis, in the writing of the manuscript, or in the decision to submit it for publication.

**Competing interests:** The authors have declared that no competing interests exist.

## Introduction

HIV/AIDS is one of the most devastating illnesses that humans have ever suffered. It remains a major public health issue impacting communities all over the world, having claimed over 40 million lives so far, and with an estimated 39 million people living with HIV globally, as at the end of 2022 [1]. Eastern and Southern Africa, an under-resourced region with significant health system constraints [2], remains the region most heavily affected, with an estimated 20.6 million people living with HIV [1,3]. Several studies have shown that depression is one of the most common comorbid conditions in people infected with HIV, with approximately 8–50% of persons living with HIV reported to have suffered from depressive disorders [4–8]. In Uganda, a recent meta-analysis found a pooled depression prevalence of 28.2% among people living with HIV/AIDS (PLWHA), nearly ten times higher than the prevalence estimates in the general population [9].

Depression in PLWHA worsens the existing disease states, as it not only affects the quality of life [10], but has been associated with poorer health outcomes such as hastening the progression to AIDS [11–13], poor adherence to HIV treatment, risky sexual behaviour and increased utilization of health facilities [14–16] and elevated risk of mortality [5,17]. Although an estimated 76% of the PLWHA globally are on antiretroviral (ARV) treatment, the majority of HIV care providers in sub-Saharan Africa do not routinely provide mental health services to address the problem of depression [3]. However, there is growing evidence of specific treatments for depression among PLWHA, associated with anti-retroviral therapy (ART) adherence and reduced HIV disease progression [18–20]. This makes a case for integrating such treatments in routine HIV care [21–23].

With an estimated 1.4 million people living with HIV/AIDS in Uganda, studies have reported exceptionally high prevalence of depressive symptoms among PLWHA, posing a major challenge in HIV care, despite the success in the scale up of anti-retroviral therapy. This consequently increases mortality [14,24]. To address the absence of mental health care in HIV programs, the Uganda National HIV and AIDS Strategic Plan (2020/21–2024/25) called for the integration of mental health and other chronic conditions in HIV care so as to further improve the quality of care and treatment. In a similar vein, the 2016 Ministry of Health policy initiative and guidelines called for the assessment and management of depression in PLWHA [25].

In response to the above calls, the Mental Health Section of the Medical Research Council (MRC)/Uganda Virus Research Institute (UVRI) & London School of Hygiene and Tropical Medicine (LSHTM) Uganda Research Unit, in partnership with the STD/AIDS Control Program of the Ministry of Health, Uganda Implemented the HIV+D intervention model. This was a 5 year project implemented in 3 districts within Uganda. The intervention model consisted of psychoeducation, Behavioural Activation, antidepressant medication and referral to mental health workers; delivered in a stepped care fashion. The therapies have previously been shown to be effective against depression in primary care settings including in HIV care [26–29].

Process evaluation, which is carried out during or after the intervention to evaluate and explain the outcomes has become a dominant part of clinical trials [30]. It focuses on how and why the intervention works or does not work in the context of the trial [31,32]. Process evaluation is critical for interpreting the outcomes of trials of complex interventions and aids in the understanding of contextual factors associated with intervention and implementation effectiveness; which can influence potential replication in different settings as well as scale up [33,34].

It is on this basis that we set out to conduct a process evaluation for the HIV+D intervention.

The aim of the study therefore was to assess and report results of the process evaluation of the HIV+D intervention. This was deemed necessary so as to understand whether the

intervention was implemented as intended and received similarly across all sites; and to understand the variation in outcomes, breadth and depth in effectiveness; thereby differentiating intervention failure from implementation failure. The process evaluation would justify any variability in the intervention outcomes and inform strategies to integrate the management of depression in HIV care.

## Methods

### Overview of the intervention

The HIV + D intervention model was based on therapies that have been shown to be effective against depressive disorder in primary care settings including in HIV care, namely: psychoeducation, behavioural activation-the Healthy Activity Program (HAP) [29] and antidepressant medications [27]. These were delivered in a stepped care format and overseen by an HIV counsellor (Lay Health Worker). Recruitment of participants in the study started on Monday 3rd May 2021 and ended on Friday 31st December 2021.

Recruitment would start with a health talk about depression, given by a trained Lay Health Worker (LHW) to PLWHs sitting in the triage area of the participating Public Health Care Facilities. Thereafter, consecutive attenders would be approached and requested to be screened for the study. The screening would be done by the trained LHW using the PHQ-2 (a two-item questionnaire that assess for 'depressed mood' and 'loss of interest in typically pleasurable activities'). Those screening positive (PHQ-2 score $\geq$ 3) would be invited for further evaluation by the trained LHW for trial eligibility and consent. The eligibility assessment would include confirmation of depressive symptoms using the PHQ-9 tool [35]. The eligible respondents would be recruited into the intervention. Those with suicidal ideation (endorsing item 9 on the PHQ-9) would immediately be referred to the supervisor for further assessment and then referral to a mental health specialist (Psychiatric Clinical Officer) for further management.

The intervention was delivered in 4 steps. Step 1: (initiation of treatment), patients with PHQ-9 scores of 10–19 would be told of their scores and offered psychoeducation by a LHW. Step 2: (management of moderate to severe cases), patients who remained symptomatic at follow up (PHQ-9 score $\geq$ 5 after 4 weeks) would be offered Behavioural Activation (BA); 4–10 bi-weekly sessions) by a LHW. Step 3: (monitoring outcomes), if after 6 sessions of BA, one still scored 10 and above on PHQ-9, they would continue BA sessions to completion and add antidepressant medication (Fluoxetine 20 mg/day for 6 months), initiated by the HIV clinician. Step 4: (referral to mental health worker), if there was no improvement (PHQ-9 $\geq$ 5) after step 3 or at eligibility assessment PHQ-9 $\geq$ 20, or if someone had a moderate to high suicide risk (based on suicide risk assessment by the supervisor), they would continue all existing treatment and be referred to a mental health worker. This is summarized in the appended CONSORT flow diagram (S1 Fig) and CONSORT checklist (S1 Checklist).

The interventions were delivered to the recruited PLWHA in the intervention sites, while those in the control sites received enhanced usual care following mental health Gap Action Plan (mhGAP) treatment guidelines. Full details of the intervention are described elsewhere in the HIV+D protocol paper, appended as supporting information (S1 Text) [36].

### Conceptual framework for implementation outcomes

The process evaluation was designed in line with the MRC guidelines for process evaluations [30]; specifically using the RE-AIM framework [37]. This is a framework that guides planning and evaluation of programs, focusing on 5 key dimensions: (i) reach (the extent to which the intervention was received by the targeted group), (ii) efficacy/effectiveness (effects of the

intervention on health outcomes), (iii) acceptability (extent to which participants found the intervention relevant, acceptable and satisfactory), (iv) implementation fidelity (extent to which the intervention was delivered as planned), (v) maintenance (extent to which the intervention can be integrated within the existing structure and sustained over time). Thus, in this paper, we report on the above 5 implementation outcomes, which formed the basis of our process evaluation. Although not part of the framework, we also report on context (environmental aspects of the intervention setting) as an additional dimension that has been acknowledged as a key dimension [32].

The specific objectives of the study were thus:

1. To assess and determine the extent to which the intervention reached the target group.

2. To document the effects of the intervention on the health outcomes in the target population.

3. To determine the extent to which the participants found the intervention relevant and satisfactory.

4. To determine the extent to which the intervention was implemented as intended across the different intervention sites.

5. To explore the perceived sustainability and integration of the intervention within the existing HIV care system

6. To describe the contexts in which the intervention was delivered and explore contextual factors that have influenced the delivery and outcomes of the intervention.

## Setting

The HIV+D program was implemented in 40 public health facilities, which were randomly selected across the 3 study districts of Wakiso and Masaka (semi-urban and rural), and Kalungu (predominantly rural). These are all in central Uganda. All the facilities were running active HIV/AIDS clinics. This process evaluation was a qualitative study, conducted in 28 of the 40 public health facilities (14 intervention sites and 14 control sites) between June 2021 and April 2022.

## Study participants and sampling

We set out to conduct the process evaluation in all the 40 health facilities and involve purposively selected participants who could provide in-depth and detailed information on the topic under investigation. These included the health workers who delivered the intervention, the study data collectors, lay health workers, persons living with HIV/AIDS and carers (treatment supporters). However, the data collection went on until we reached saturation and further data collection was deemed redundant; after covering 28 health facilities, with a total of 82 participants.

## Data collection

Data was collected through key informant interviews (KIIs) and focus group discussions (FGDs) to understand the context and process, characteristics of the intervention, effectiveness, facilitating factors and challenges from the perspective of different stakeholders. An interview guide with items covering these aspects was developed for this purpose. A total of 58 KIIs and 6 FGDs were conducted with the different stakeholders, as summarized in Table 1

**Table 1. Interviews and FGDs conducted.**

|   | Category of respondents | No. of Key Informant Interviews | No. of Focus Group Discussions |
|---|---|---|---|
| 1 | Health workers and supervisors | 11 | 2 |
| 2 | Lay health workers | 3 | 1 |
| 3 | Data collectors | 7 | 0 |
| 4 | Persons living with HIV (clients) | 25 | 3 |
| 5 | Carers and treatment supporters | 10 | 0 |
| 6 | Expert clients | 2 | 0 |

below. All interviews and FGDs were audio-recorded. The interviews with health workers and data collectors were conducted in English, while those with the LHWs, PLWHA and carers were conducted in Luganda (the local language) and translated during the transcription. The transcription and translation were done by the first author and one of the research assistants, both being very conversant with the 2 languages, and having strong experience in qualitative data collection methods.

## Data analysis

After the transcription, conventional content analysis was done by the 2 researchers who have expertise in qualitative research and data analysis, to ensure rigour. They subjectively interpreted the content of the transcripts through coding and identifying themes [38,39]. The initial coding of the transcripts was undertaken by the first author, guided by predefined themes and sub-themes derived from the study objectives. These were amended along the way based on the transcription data. The coded transcripts were shared with the first co-author for common interpretation, improvement and consensus. A data analysis matrix sheet was then used to generate a concise summary of the key results.

## Ethical considerations

Ethical approval was obtained from the UVRI Research and Ethics Committee (07th April, 2020), the LSHTM Ethics Committee (17th September, 2020) and the Uganda National Council for Science and Technology (ethical clearance number: HS645ES) (07th July, 2020). All study participants gave written informed consent to participate in the study.

# Results

The results are presented below, under 5 themes: effectiveness, acceptability, implementation fidelity, maintenance (sustainability) and contextual factors that affected the intervention. Given the study design, it was not possible to meaningfully assess "reach".

## Acceptability

The key implementers for the program at the primary health care facilities included health workers (clinical staff), lay health workers (members of the Village Health Teams), data collectors. Some of the respondents could not give a very precise description of the program, but indicated that it was a program for treating depressed PLWHA. The non-health care professionals (LHWs and data collectors) termed it as "a program for helping PLWHA who were depressed to deal with their depression, as part of their HIV treatment program, so as to live positively."

It emerged that the counselling provided in the routine HIV clinics mostly focused on the importance of adherence to treatment (taking ARVs), rarely addressing the clients' psychosocial concerns and depressive feelings. However, the PLWHA reiterated that under the HIV+D intervention, they received a kind of counselling they had never received before, which helped them dealing with their psychosocial problems; although many did not know that depression is an illness; and its term in the local language was not familiar to them.

Aspects of acceptability included the respondents' level of interest in the HIV+D program and perceptions about its relevance and usefulness as well as the level of satisfaction. The intervention and its content met the respondents' approval, and was generally considered relevant and acceptable. The participating health workers and supervisors commented on the appropriateness of the intervention, affirming that the program addressed a real need of the target population and was much needed, given the agony, health and socio-economic challenges associated with being HIV positive. They further reported having a sense of contentment that the intervention provided for helping the depressed PLWHA more meaningfully. On the other hand, some of the PLWHA admitted that at the time of this intervention, they were still struggling to come to terms with their HIV status; and the program had greatly helped them in this aspect. There were reports of increased awareness and recognition of depression as a common health problem among PLWHA, for which they ought to seek help. One PLWHA stated:

"...Before, it was like I was in my own world, because of my HIV status. I had nothing to do to earn some money. It also seems there was another disease, inside me and I had so many thoughts...so many questions...||...I was taking the medicine (ARVs) but it wasn't working for me... and that was mentioned by the health workers, that if you have so many thoughts, the medicine you're taking will not work for you. Personally, I was depressed and doing badly"

(KII 40, PLWHA, Busawamanze HC III).

A health worker affirmed:

"... at first they did not even know that depression is an illness, some would tell us: 'Nurse I did not know that to feel like this, it's an illness. Someone can have problems, move with them every day but does not know that it is an illness'. So patients got to know that there is a solution to those problems and one does not have to be depressed."

(FGD 06, health workers, Busawamanze HCIII).

The respondents across the different health facilities gave reports indicating high levels of patient satisfaction with the service. For example, one supervisor said:

"...Someone comes and say "nurse you have helped me a lot. Those things sessions that been through have helped me understand myself, I feel better. It means that the patient is appreciative and feels helped, I am satisfied and feel it"

(Respondent 1, FGD 02, Supervisors).

Patient satisfaction was further confirmed by the fact that some of the PLWHA and carers identified and referred fellow PLWHA in their communities, whom they believed to be depressed, for recruitment into the program. Similarly, the health workers at the facilities offering the intervention reported an increase in the number of clients seeking treatment at these facilities, attracted by the program.

## Implementation fidelity

According to the protocol, the implementation would start with a health talk on depression, delivered by a lay health worker, followed by screening for depressive symptoms. The screen positive would be further subjected to an eligibility assessment to confirm the depressive symptoms before consenting for enrollment. In their accounts, most respondents affirmed that screening for depressive symptoms would be done using the PHQ-9, followed by testing for eligibility before recruitment into the program. The screen positives would initially receive psychoeducation and those whose symptoms persisted be recruited to receive Behavioural Activation therapy, which was the main intervention for the program. These would attend bi-weekly BA sessions until full remission of the symptoms. At the control sites, they would assess for depression and any other physical condition such as high blood pressure and then give medication accordingly; though not giving the clients much time, as was the case at intervention sites. However, there were some variations and confusion in some of the respondents' description of the intervention design and components, as they seemed to confuse the procedure and order of the steps and to confuse the initial health talk that would be given, with psychoeducation which was offered as part of the therapy.

It was noted that at all health facilities, the implementers duly adhered to the intervention protocol, conducting the activities according to the plan, without major program adaptations or alterations of any critical components relating to the content, activities or delivery of the program. The process evaluation thus revealed strong implementation fidelity.

## Effect of the intervention on health outcomes of the target population

The intervention was said to have been delivered with excellent facilitation skills, enthusiasm and positive attitude across the implementation sites, which resulted in positive outcomes. It was noted that the intervention resulted in increased awareness and recognition of depression as a health problem, awakening the health workers' awareness to focus beyond presenting complaints and symptoms while handling patients.

Apparently many of the PLWHA with depressive symptoms did not know that they were having a health problem that warranted clinical intervention. The health workers on the other hand confessed that they were previously not offering adequate help because they were less confident in assessing for depression, and did not give it consideration, as reflected in the quote below:

"...*we the health workers did not know these symptoms. We were not assessing the clients for depression because we did not have that knowledge...and we did not know it was important anyway*"

(KII 46, Health facility manager/supervisor)

The intervention was reported to have resulted in increased competence and confidence to detect and treat depression associated with HIV or any other causes. Similarly, the health workers appreciated the importance of giving the patients more time and conducting a detailed assessment to inquire about patients' psychosocial problems.

"...*What I observed in this study if you have not talked to the person you cannot know what they are going through but when you talk with them at length, you discover that people have problems. You watch the person coming in to the clinical room as usual things seem normal but if you start a conversation with that person and give him/her time, you find that they are depressed*"

(Respondent 2, FGD 01, study supervisors).

Another health worker echoed:

*". . .even us health workers in ART clinic, we were not emphasizing depression treatment so much. We would only ask them about how many tablets the patient is left with. And if one is not the taking medicine, we only asked "why", but we could not go deep to find out whether the client is depressed or not"*

(KII 44, Health worker)

It was reported that by design, the intervention strengthened the bond between the health workers and PLWHA, resulting in improved Patient-Health worker relationship, empathy and compassion. The improved relationship was said to have contributed to increased disclosure by the PLWHA to the health workers as well as the family members, for those who were living in denial or struggling to keep their HIV status a secret, resulting in increased adherence to treatment. This is reflected in the voices below:

*". . .Before [the intervention], we would come to the facility but never got a chance to sit with a health worker to talk like we are doing now. O.k the nurse would ask whether you have anything bothering you, whether you are taking the medicine. . ..but doing so in a rush. You could see that she is after clearing the long queue. So, it would be hard to disclose important information"*

(KII 054, PLWHA, Mpugwe HCIII).

One health worker also affirmed:

*". . .So, because of the good relationship we have now, they don't miss on their appointments. And sometimes a client comes just to greet you. But before [the intervention], that could not happen for a patient to come and say "nurse I have this problem" the moment you give them medicine, they would just walk away. Sometimes this patient may be going through hardships and you the health worker can never know about it unless the patient discloses"*

(Respondent 01, FGD 06, PHC workers)

The PLWHA anticipated that the improved relationship they developed with the health workers in the course of the program would be an opportunity to leverage on and serve as a potential buffer against likely psychosocial problems long after the project.

The health workers at various health facilities reported an increase in prescription and consumption of the anti-depressant medication, which would previously expire at some of the facilities. Consequently, there was a reduction in the depressive symptoms among the beneficiaries. This was further exemplified by the significant drop in the clients' PHQ-9 scores on the subsequent visits.

One of the remarkable outcomes reported was the improvement in adherence to treatment and positive living. Some of the PLWHA confessed that they previously deliberately defaulted on treatment and could not see the point in continuing with medication that would not lead to absolute healing. However, they had resumed taking their ARVs following the intervention.

*". . .before this program, I would come pick the medicine, but I was not taking it. . .would just keep it home waiting for death. . ..||. . .when I came and had a health talk with the health worker, I went home picked the tins of medicine and brought them back to the health facility. I confessed that I was not taking the medicine. Actually some tins had expired,*

*so they were thrown away. But now, am taking the medicine and have not skipped even a day*"

(KII 054, PLWHA, Mpugwe HCIII).

It emerged that for some PLWHA, defaulting on taking their ARV medication was driven by their suicidal ideation, wishing they could die in the process. Indeed, some of them shared testimonies indicating how they had contemplated suicide at some point, because of their HIV status and how they had deliberately defaulted on ARV medication, hoping it would be an easy gateway to death without having to use the more obvious means of committing suicide. There were reports of several PLWHA who were previously very depressed and contemplating suicide, but had improved tremendously following the intervention.

Another key outcome proving the efficacy of the intervention relates to the reported notice-able increase in viral load suppression among PLWHA at several health facilities. The health workers reported on several cases of PLWHA who had failed to suppress the viral load despite taking their ARVs for some time, but were able to achieve the suppression in the course of this intervention.

The respondents also cited some noteworthy unintended effects of the intervention. One such effect was the gain in terms of capacity building for the lay health workers who were trained in delivering BA intervention. They affirmed that delivering the intervention was in some way a learning experience for them as well, having acquired more knowledge and skills in counselling in the process, which could potentially enable sustainability of the intervention.

One such effect was the reported improvement in the welfare of some PLWHA, attributable to the small financial support they received as transport refund whenever they attended ther-apy sessions. One respondent revealed:

"*. . .There are some two women in their 50s who said they had given up on life. One of them said 'this money they give us, now I was able to buy a hoe or shoes. Because previously, I was walking bare-footed. I was even able to buy cups and plates for the home. Previously, one would have to take tea and wash the cup so that another one can also take. We did not have enough cups and plates. We would get maize from the garden, cook it and put on banana leaves and take water as soup. But you have helped us a lot. Now everyone has a plate and a cup'. So, you realize that the program has helped this person financially*".

(KII 028, Expert Client, Kyanamukaaka HCIV).

## Sustainability

The health workers were optimistic that the intervention could be adopted and incorporated into the existing HIV/AIDS care system. However, they expressed uncertainty over continuity of the intervention and sustainability of the gains, due to challenges such as understaffing and resource constraints. The PLWHA and carers too were skeptical that the quality of service would remain the same under the mainstream health care system, given the resource require-ments. They thus maintained that adoption of such an intervention would necessitate scaling up of the staffing in light of the fact that the intervention entails health workers giving patients adequate time. The health workers further expressed willingness to continue delivering the intervention even after the end of the trial.

## Contextual factors

The context and mode of operation at all the health facilities was generally the same, save for some difference in the staffing levels. However, several contextual factors relating to health worker's attitude and skills, resources, organizational norms etc affected the implementation and the outcomes.

The respondents affirmed that the intervention addressed an actual need and was long overdue. The PLWHA and carers specifically commented on the humility, compassion and the positive attitude of the health workers who delivered the intervention as key facilitating factors that enabled smooth running of the intervention. The warm patient-health worker relationship has been key in fostering adherence and recovery. This was further backed by teamwork and a good working relationship that the implementers exhibited.

According to the health workers, the training they received to deliver the intervention was thorough, backed by close supervision by other project staff, which enabled them to appreciate the intervention well. In addition, they showed commitment towards the work, and were motivated by the incentive; thereby giving the program adequate time.

The transport refund given to the PLWHA recruited into therapy was also cited as a major motivating factor for patients' attendance. Apparently, many of them previously did not show interest to turn up for the counseling sessions in the routine HIV care. Relatedly, it was reported that the beneficiaries of the intervention (PLWHA) had high expectations, in terms of material support at the time of recruitment. This was believed to have influenced their compliance and consistence in attending the therapy sessions.

There were also a few contextual factors that negatively affected implementation. Most notable of these was the nearly 2 years of Covid-19 lockdown characterized by high levels of anxiety, restriction of movement and disruption of work and people's lives. Implementation was mostly around that timing, and necessitated conducting some of the therapy sessions on phone.

Like in any study setting, there was a limit on the number of clients to recruit at the health facilities, to a maximum of 30 PLWHA. According to some of the respondents, the limit meant leaving out some other potentially depressed clients who would actually have benefited from the service.

One major challenge in the implementation of the intervention was to do with the screening and testing clients for eligibility. It emerged that by design, the program involved too much paperwork right from assessment of individuals for eligibility, recruitment into therapy and other processes. This was considered to be practically cumbersome if the intervention is to be replicated or rolled out under normal circumstances (outside a research study context).

> "... we talked about the long questionnaires... time. Some of the patients would get impatient. And our patients would always want their visits to coincide with the day for picking their drugs. Time was a major concern"

(FGD 02, Health workers, Bukakata HC III)

Related to the above, there was a possible risk of some of the PLWHA answering questions just for the sake, when already tired of the so many questions, thereby giving wrong information. Furthermore, some of the lay health workers and expert clients thought the idea of using screening tools in recruiting clients into therapy was superficial, arguing that all PLWHA have psychosocial problems and ideally should benefit from the service.

## Discussion

The HIV+D process evaluation findings indicated that the intervention reached the target population and was generally implemented and received as intended. Most stakeholders described the intervention as "a program for helping PLWHA who were depressed to deal with their depression"; an indicator that they were clear about the goal of the intervention and the target population. Importantly, the PLWHA acknowledged the fact that they received a form of counselling quite different from the usual HIV counselling delivered in the HIV clinics. The fact that most of them initially did not consider depression to be an illness is of particular concern, as it has implications on the likelihood of seeking help.

The intervention received approval from the implementers (health workers) as well as the recipients (PLWHA and carers); who considered it to be acceptable and very necessary. Importantly, several respondents, including health workers believed that all PLWHA would ideally benefit from the intervention in light of the fact that acquiring HIV can be a serious psychological trauma, predisposing one to psychological distress and different mental disorders [40,41]. This was further confirmed by health workers who cited several examples of PLWHA who earlier on seemed to be doing well, but turned out to be harboring serious distressing problems which were unearthed during the therapy sessions, much to the surprise of the health workers. This could be one reason why at the time of this evaluation, most respondents wished for continuity of the program and expressed disappointment to learn of its impending closure.

Fidelity was a function of the implementers and encompassed both the quality and quantity of delivery [42]. As earlier noted, not all respondents could accurately describe the intervention and give a perfect narration of how it was implemented from the onset. However, their summative description indicated that the health workers endeavoured to comply with the procedure and activities as set out in the intervention protocol, with no significant adaptations.

It is worth noting that both the implementers and recipients of the intervention expressed concern over the much paperwork involved, especially in the screening and recruitment process. While this was a procedural requirement as per the design of the intervention, and also important for scientific rigor (for the research aspect), it may be rather cumbersome and not practically viable in the normal clinical setting. Indeed, this emerged as one of the challenges encountered during the implementation, and would need to be considered critically while making arrangements for scaling up the intervention.

Some of the health workers cited some examples of PLWHA who disclosed during the therapy sessions that they had earlier contemplated suicide because of their HIV status, but had not responded affirmatively to the suicide item during the assessment. Several earlier studies have asserted that suicidal behaviours are much more pronounced in PLWHA [43–46], citing the relationship between HIV-related stigma, depression and suicidal ideation among PLWHA [47]. This therefore speaks to the likely concealment of the suicidal ideation during the screening and the need for thorough assessment so as to respond and intervene appropriately. One of the striking findings was the revelation by some of the PLWHA, that they deliberately defaulted on their ARV medication, with the ulterior motive of dying in the process. Several studies have found PLWHA to have a higher likelihood of suicidal ideation due to a significant burden of disease they have to contend; often mediated by several psychosocial variables such as high perceived stigma, low self-esteem, social support and resilience [48–50].

The findings further alluded to effectiveness of the intervention. According to the respondents, the intervention resulted in several positive treatment outcomes (both direct and indirect), demonstrating the importance of giving PLWHA in therapy adequate time to assess their psychosocial wellbeing and understand their plight. The intervention was thus

considered a success despite being implemented during the difficult times of Covid-19 lockdown, associated with numerous challenges causing distress to communities. The observed efficacy is in support of other earlier studies that have reported on the effectiveness of Behaviour Activation therapy in treating depression, even though little is known about its working mechanisms [51,52]. The beneficiaries appreciated the intervention, as it apparently gave them something beyond what they ordinarily received under the usual HIV care system; illustrating the effectiveness of psychological treatments for depression.

Several contextual factors were reported to have affected the implementation of the intervention. It should be noted that both the implementers (health workers) and the beneficiaries (PLWHA and their carers) overemphasized the facilitation in terms of transport refund as having been key to their participation and consistence. Given the high poverty levels, some would even endure walking the long distances, so as to save the money and support their financial needs.

This seems to suggest that many of them were mostly driven by some extrinsic motivation, instead of their underlying condition; implying that their attendance and consistence would probably have been difficult if there was not such facilitation, as a motivating factor. This casts doubt on the sustainability of the intervention at health facility level under the current health system, characterized by inadequate staffing and resource constraints. Nevertheless, the health system ought to devise strategies for scaling up or integrating the intervention in the mainstream HIV care in light of its potential positive effects.

## Conclusion

In light of the above findings, we reasonably conclude that the intervention was generally implemented as planned, and considered acceptable and satisfactory by the implementers and beneficiaries. All respondents concurred with the fact that the intervention was beneficial and feasible; calling for its integration into the mainstream HIV/AIDS care system.

### Lessons learnt and recommendations

Some of the lessons learnt in the course of the implementation, which could potentially inform the scale up of the intervention include:

1. The importance of a warm patient-health worker relationship and its potential impact on treatment outcomes.

2. The importance of supervising the implementers very closely when rolling out such an intervention

3. The role played by stigma and discrimination in non-adherence to treatment and loss to follow up among PLWHA.

4. Poverty is a strong mediating factor leading to depression among the PLWHA.

    In light of the above, we make the following recommendations:

1. The need to prioritize training of all health workers involved in HIV care in mental healthcare, to be able to assess for, detect and manage the less obvious psychosocial problems and their manifestation among PLWHA.

2. While there were reports of stigma and discrimination against the PLWHA, the participants' revelations apparently pointed towards more of self-perceived (internalizing stigma). Interventions for PLWHA should therefore not overlook this problem.

3. The need for special consideration of the PLWHA who are socio-economically disadvantaged, prioritizing them in economic support programs such as the Social Assistance Grants for Empowerment (SAGE) program.

## Limitations of the study

The intervention had to go on during the COVID-19 pandemic. The lock-down that ensued and the associated restrictions certainly reduced activity and participation rates; implying that the overall impact of the intervention could have been more significant had it not been the disruptions by the COVID-19 pandemic. However, the process evaluation did not assess for the effect of the pandemic on the programme.

One other limitation of the study is the fact that we relied on qualitative measurements, collecting data from a relatively small sample of individuals, which may affect the generalization of the findings. Furthermore, we did not do any quantitative assessment of the intervention outputs; making it hard to assess and demonstrate rigour.

## Supporting information

**S1 Checklist. SPIRIT checklist (CONSORT checklist).** Standards of how the HIV+D trial was designed, analyzed and interpreted.
(DOCX)

**S1 Fig. HIV+D Consort flow diagram.** Visual representation of the steps involved (progress and phases) for the HIV+D trial.
(TIF)

**S1 Text. HIV+D protocol paper.** Article about the HIV+D study, highlighting the rationale and objectives of the study, methodology, data management and analysis.
(PDF)

## Author Contributions

**Conceptualization:** Joshua Ssebunnya, Hafsa Sentongo, Pontiano Kaleebu, Vikram Patel, Eugene Kinyanda.

**Formal analysis:** Joshua Ssebunnya.

**Funding acquisition:** Pontiano Kaleebu, Eugene Kinyanda.

**Investigation:** Christine Tusiime.

**Methodology:** Joshua Ssebunnya, Richard Mpango, Leticia Kyohangirwe.

**Project administration:** Richard Mpango, Eugene Kinyanda.

**Resources:** Pontiano Kaleebu, Vikram Patel.

**Supervision:** Leticia Kyohangirwe, Christine Tusiime, Hafsa Sentongo, Vikram Patel, Eugene Kinyanda.

**Validation:** Rutakumwa Rwamahe, Eugene Kinyanda.

**Writing – original draft:** Joshua Ssebunnya.

**Writing – review & editing:** Rutakumwa Rwamahe, Leticia Kyohangirwe, Christine Tusiime.

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

Methods Process Evaluation: Integrating Depression Treatment Into HIV Care in Malawi. Glob Health
Sci Pract [Internet]. 2021 Sep 30. [cited 2024 January 4]; 9(3):611–625. Available from: https://www.
ncbi.nlm.nih.gov/pmc/articles/PMC8514021/ https://doi.org/10.9745/GHSP-D-20-00607

22. Kulisewa K, Stockton MA, Hosseinipour MC, Gaynes BN, Mphonda S, Udedi MM, et al. The Role of
Depression Screening and Treatment in Achieving the UNAIDS 90-90-90 Goals in Sub-Saharan
Africa. AIDS Behav [Internet]. 2019 July 17. [cited 2023 Sep]; 23(Suppl 2):153–161. Available from:
https://www.ncbi.nlm.nih.gov/pmc/articles/PMC6773678/ https://doi.org/10.1007/s10461-019-
02593-7

23. Mills JC, Pence BW, Todd JV, Bengtson AM, Breger TL, Edmonds A, et al. Cumulative Burden of
Depression and All-Cause Mortality in Women Living With Human Immunodeficiency Virus. Clin Infect
Dis [Internet]. 2018 March 30. [cited 2024 January 3]; 67:1575–81. Available from: https://doi.org/10.
1093/cid/ciy264 PMID: 29618020

24. Uganda AIDS Commission [Internet]. Kampala: Uganda AIDS Commission; 2022. Facts on HIV and
AIDS in Uganda, 2022 [cited 2023 October 11]. https://uac.go.ug/media/attachments/2023/01/12/hiv-
aids-factsheet-2022.pdf.

25. Uganda AIDS Commission [Internet]. Kampala: Uganda AIDS Commission; 2015. The Uganda
National HIV and AIDS Strategic Plan 2015/2016–2019/2020 [cited 2023 June 19]. http://library.health.
go.ug/publications/service-delivery-diseases-control-prevention-communicable-diseases/hivaids/
national-h-1.

26. Abbas Q, Nisa M, Khan MU, Anwar N, Aljhani S, Ramzan Z, et al. Brief cognitive behavior therapy for
stigmatization, depression, quality of life, social support and adherence to treatment among patients
with HIV/AIDS: a randomized control trial. BMC Psychiatry [Internet]. 2023 July 25. [cited 2024 January
4]; 23, 539. Available from: https://doi.org/10.1186/s12888-023-05013-2 PMID: 37491185

27. El-Halabi S, Cooper DH, Cha DS, Rosenblat JD, Gill B, Rodrigues NB, et al. The effects of antidepres-
sant medications on antiretroviral treatment adherence in HIV-positive individuals with depression.
Journal of Affective Disorders [Internet]. 2022 Mar 1. [cited 2024 January 3]; 300:219–25. Available
from https://pubmed.ncbi.nlm.nih.gov/34952118/ https://doi.org/10.1016/j.jad.2021.12.083

28. Qin K, Zeng J, Liu L, Cai Y. Effects of cognitive behavioral therapy on improving depressive symptoms
and increasing adherence to antiretroviral medication in people with HIV. Front. Psychiatry [Internet].
2022 Nov 9 [cited 2024 January 4]. Available from: https://www.frontiersin.org/journals/psychiatry/
articles/10.3389/fpsyt.2022.990994/full PMID: 36440403.

29. Chowdhary N, Anand A, Dimidjian S, Shinde S, Weobong B, Balaji M, et al. The Healthy Activity Pro-
gram lay counsellor delivered treatment for severe depression in India: systematic development and
randomised evaluation. The British journal of psychiatry. 2016; 208(4):381–8. https://doi.org/10.1192/
bjp.bp.114.161075 PMID: 26494875

30. Moore GF, Audrey S, Barker M, Bond L, Bonell C, Hardeman W, et al. Process evaluation of complex
interventions: Medical Research Council guidance. BMJ [Internet]. 2015 Mar 19. [cited 2023 July 15];
350:h1258. Available from: https://pubmed.ncbi.nlm.nih.gov/25791983/ https://doi.org/10.1136/bmj.
h1258

31. French C, Pinnock H, Forbes G, Skene I, Taylor SJC. Process evaluation within pragmatic randomised
controlled trials: what is it, why is it done, and can we find it?—a systematic review. Trials [Internet].

2020 November 9. [cited 2023 July 23]; 21, 916. Available from: https://doi.org/10.1186/s13063-020-04762-9 PMID: 33168067

32. Grant A, Bugge C, Wells M. Designing process evaluations using case study to explore the context of complex interventions evaluated in trials. Trials [Internet]. 2020 November 27. [cited 2023 July 22]; 21, 982. Available from: https://trialsjournal.biomedcentral.com/articles/10.1186/s13063-020-04880-4. PMID: 33246496

33. Limbani F, Goudge J, Joshi R, Maar MA, Miranda JJ, Oldenburg B, et al. Process evaluation in the field: global learnings from seven implementation research hypertension projects in low-and middle-income countries. BMC Public Health [Internet]. 2019 July 16. [cited 2023 July 25]; 19, 953. Available from: https://doi.org/10.1186/s12889-019-7261-8 PMID: 31340828

34. Hickey G., McGilloway S., Furlong M, Leckey Y, Bywater T, Donnelly M. Understanding the implementation and effectiveness of a group-based early parenting intervention: a process evaluation protocol. BMC Health Serv Res [Internet]. 2016 September 15. [cited 2023 July 24]; 16, 490. Available from: https://doi.org/10.1186/s12913-016-1737-3 PMID: 27633777

35. Kroenke K, Spitzer RL. The PHQ-9: a new depression diagnostic and severity measure. Psychiatric Annals [Internet]. 2001 September 16. [cited 2023 July 20]; 32(9):509–15. Available from: https://pubmed.ncbi.nlm.nih.gov/11556941/.

36. Kinyanda E, Kyohangirwe L, Mpango RS, Tusiime C, Ssebunnya J, Katumba K, et al. Effectiveness and cost-effectiveness of integrating the management of depression into routine HIV Care in Uganda (the HIV +D trial): A protocol for a cluster-randomised trial. Int J Ment Health Syst [Internet]. 2021 May 12. [cited 2023 July 12]; 15, 45. Available from: https://ijmhs.biomedcentral.com/articles/10.1186/s13033-021-00469-9.

37. Kwan BM, McGinnes HL, Ory MG, Estabrooks PA, Waxmonsky JA, Glasgow RE. RE-AIM in the Real World: Use of the RE-AIM Framework for Program Planning and Evaluation in Clinical and Community Settings. Front Public Health [Internet]. 2019 Nov 22. [cited 2023 August 12]; 7:345. Available from: https://www.ncbi.nlm.nih.gov/pmc/articles/PMC6883916/ https://doi.org/10.3389/fpubh.2019.00345

38. Assarroudi A, Heshmati Nabavi F, Armat MR, Ebadi A, Vaismoradi M. Directed qualitative content analysis: the description and elaboration of its underpinning methods and data analysis process. J Res Nurs [Internet]. 2018 Feb. [cited 2023 July 20]; 23(1):42–55. Available from: https://pubmed.ncbi.nlm.nih.gov/34394406/ Epub 2018 Jan 10. https://doi.org/10.1177/1744987117741667

39. Hsieh HF, Shannon SE. Three approaches to qualitative content analysis, Qual Health Res [Internet]. 2005 November. [cited 2023 July 15]; 15, 1277–1288. Available from: https://doi.org/10.1177/1049732305276687 PMID: 16204405.

40. Moges NA, Adesina OA, Okunlola MA, Berhane Y, Akinyemi JO. Psychological Distress and Its Correlates among Newly Diagnosed People Living with HIV in Northwest Ethiopia: Ordinal Logistic Regression Analyses. Infect Dis (Auckl) [Internet]. 2021 Feb 14. [cited 2023 December 30]. Available from: https://www.ncbi.nlm.nih.gov/pmc/articles/PMC7890707/ https://doi.org/10.1177/1178633721994598

41. Motumma A, Negesa L, Hunduma G, Abdeta T. Prevalence and associated factors of common mental disorders among adult patients attending HIV follow up service in Harar town, Eastern Ethiopia: a cross-sectional study. BMC Psychol [Internet]. 2019 February 22. [cited 2023 September 23]; 7, 11. Available from: https://bmcpsychology.biomedcentral.com/articles/10.1186/s40359-019-0281-4 PMID: 30795804

42. An M, Dusing SC, Harbourne RT, Sheridan SM, START-Play Consortium. What Really Works in Intervention? Using Fidelity Measures to Support Optimal Outcomes, Physical Therapy [Internet]. 2020 January 16. [cited 2023 August 20]; 5,757–765. Available from: https://academic.oup.com/ptj/article/100/5/757/5707307.

43. Necho M, Tsehay M, Zenebe Y. Suicidal ideation, attempt, and its associated factors among HIV/AIDS patients in Africa: a systematic review and meta-analysis study. Int J Ment Health Syst [Internet]. 2021 Jan 23. [cited 2024 January 5]; 15(1):13. Available from: https://www.ncbi.nlm.nih.gov/pmc/articles/PMC7825170/ https://doi.org/10.1186/s13033-021-00437-3 PMID: 33485362.

44. Pelton M, Ciarletta M, Wisnousky H, Wisnousky H, Lazzara N, Manglani M, et al. Rates and risk factors for suicidal ideation, suicide attempts and suicide deaths in persons with HIV: a systematic review and meta-analysis General Psychiatry [Internet]. 2021 April. [cited 2023 December 29]; 34:e100247. Available from: https://gpsych.bmj.com/content/34/2/e100247 PMID: 33912798

45. Gebreegziabhier KG, Kassaw DC. Lifetime Prevalence and Determinants of Suicidal Ideation and Attempt Among All Patients Living with HIV/AIDS in Hiwot Fana Specialized Hospital, Harar, Ethiopia, 2020. HIV AIDS (Auckl) [Internet]. 2020 Aug 6. [cited 2024 January 6]; 12:331–339.Available from: https://www.ncbi.nlm.nih.gov/pmc/articles/PMC7418163/ https://doi.org/10.2147/HIV.S257502

46. Fu J, Chen X, Dai Z, Huang Y, Xiao W, Wang H, et al. HIV-related stigma, depression and suicidal ideation among HIV-positive MSM in China: a moderated mediation model. BMC Public Health [Internet].

2023 October 27. [cited 2024 January 6]; 23, 2117.Available from: https://doi.org/10.1186/s12889-023-17047-y PMID: 37891525

47. Yu Y, Luo B, Qin L, Gong H, Chen Y. Suicidal ideation of people living with HIV and its relations to depression, anxiety and social support. BMC Psychol [Internet]. 2023 May 16. [cited 2024 January 7]; 11, 159. Available from: https://doi.org/10.1186/s40359-023-01177-4 PMID: 37194090

48. Gizachew KD, Chekol YA, Basha EA, Mamuye SA, Wubetu AD. Suicidal ideation and attempt among people living with HIV/AIDS in selected public hospitals: Central Ethiopia. Ann Gen Psychiatry [Internet]. 2021 February 19. [cited 2023 September 21]; 20,15 (2021).Available from: https://doi.org/10.1186/s12991-021-00335-5 PMID: 33608017

49. Wang W, Xiao C, Yao X, Yang Y, Yan H, Li S. Psychosocial health and suicidal ideation among people living with HIV/AIDS: A cross-sectional study in Nanjing, China. PLoS One [Internet]. 2018 Feb 22. [cited 2023 December 30]; 13(2):e0192940. Available from: https://journals.plos.org/plosone/article?id=10.1371/journal.pone.0192940

50. Janssen NP, Hendriks GJ, Baranelli CT, Lucassen P, Oude Voshaar R, et al. How Does Behavioural Activation Work? A Systematic Review of the Evidence on Potential Mediators. Psychother Psychosom [Internet]. 2021; 90(2):85–93. Epub 2020 Sep 8. https://doi.org/10.1159/000509820 PMID: 32898847.

51. Richards DA, Rhodes S, Ekers D, McMillan D, Taylor RS, Byford S,. Cost and Outcome of BehaviouRal Activation (COBRA): a randomised controlled trial of behavioural activation versus cognitive-behavioural therapy for depression. Health Technol Assess [Internet]. 2017 August 27. [cited 2023 September 22]; 21(46):1–366. Available from: https://pubmed.ncbi.nlm.nih.gov/27461440/ https://doi.org/10.3310/hta21460

52. Uphoff E, Ekers D, Dawson S, Richards D, Churchill R. Behavioural activation therapies for depression in adults. Cochrane Database Syst Rev [Internet]. 2019 Apr 12. [cited 2023 September 23]; 4: CD013305. Available from: https://www.ncbi.nlm.nih.gov/pmc/articles/PMC6461437/ https://doi.org/10.1002/14651858.CD013305

