## [Decision Letter · Decision Letter 0]

15 Jan 2024

PMEN-D-23-00031

Process evaluation of the HIV+D intervention for integrating the management of depression in routine HIV care in Uganda

PLOS Mental Health

Dear Dr. Ssebunnya_Joshua,

Thank you for submitting your manuscript to PLOS Mental Health. After careful consideration, we feel that it has merit but does not fully meet PLOS Mental Health’s publication criteria as it currently stands. Therefore, we invite you to submit a revised version of the manuscript that addresses the points raised during the review process.

We look forward to receiving your revised manuscript.

Kind regards,

Justus Uchenna Onu, MBBS, FWACP, FMCPsych

Academic Editor

PLOS Mental Health

Journal Requirements:

https://journals.plos.org/mentalhealth/s/figures 

https://journals.plos.org/mentalhealth/s/figures#loc-file-requirements 

Additional Editor Comments (if provided):

Your manuscript has been reviewed with minor corrections.

However, the academic editor has also raised some concerns shown below.

Kindly address them.

Editorial Comments

In addition to the comments raised by the reviewers, kindly address these comments from the editor.

• Too many abbreviations without prior definition e.g., MRC, mhGAP

• How was the participating sites selected? Any selection criteria?

• I doubt if “Reach” can easily be assessed in this design. I think “Reach” should be the number of persons with HIV who received screening for depression or those who received the intervention for depression. I believe, the authors can easily assess acceptability, fidelity, sustainment and the context.

• The acceptability was reported only at the client/patient level. What about acceptability at the level of health care providers and the participating institutions?

• The acceptability dimension also measures satisfaction to intervention as mentioned by the authors. In addition to the qualitative method, addition of a validated patient satisfactiom questionnaire would have made it more rigorous.

• In the context of implementation, the authors highlighted the barriers to to successful intervention but were silent about the qualities of the environment or the intervention that facilitated the success of the program.

• What the limitations of this study?

Thank you

Reviewers' comments:

Reviewer's Responses to Questions

**Comments to the Author**

1. Does this manuscript meet PLOS Mental Health’s publication criteria? Is the manuscript technically sound, and do the data support the conclusions? The manuscript must describe methodologically and ethically rigorous research with conclusions that are appropriately drawn based on the data presented.

Reviewer #1: Yes

Reviewer #2: Yes

2. Has the statistical analysis been performed appropriately and rigorously?

Reviewer #1: N/A

Reviewer #2: Yes

3. Have the authors made all data underlying the findings in their manuscript fully available (please refer to the Data Availability Statement at the start of the manuscript PDF file)?

Reviewer #1: No

Reviewer #2: Yes

4. Is the manuscript presented in an intelligible fashion and written in standard English?

Reviewer #1: Yes

Reviewer #2: Yes

5. Review Comments to the Author

Reviewer #1: I would suggest 2 minor changes to the article: 1) it would be great to have more extended conclusions, so they represent all the major findings, not just the policy recommendations, 2) the data that can be made available, must be made available, there are formats for the qualitative data availability; please consider this. Otherwise the article is great :-)

Reviewer #2: Method:

1. what is DD? please write in full before using the acronym.

2. Fluoxetine 20mg/day/6months: Did every patient on this regimen respond and tolerate the medication for the 6/12?

3. Who precisely do you refer to as Mental Health Worker? Please, specify.

Result:

1. in table 1, No. of interviewees (not 'interviews') you mean?

Ethical consideration:

1. state the date the ethical approval was issued.

Reference:

Majority of your references cited were old (more than 5 years of publication). You may update them with newer studies as much as possible.

Goodluck!

6. PLOS authors have the option to publish the peer review history of their article (what does this mean?). If published, this will include your full peer review and any attached files.

**Do you want your identity to be public for this peer review?** For information about this choice, including consent withdrawal, please see our Privacy Policy.

Reviewer #1: No

Reviewer #2: **Yes: **Dr. Sunday O ORIJI

---

## [Editor Report · Decision Letter 1]

22 Mar 2024

Process evaluation of the HIV+D intervention for integrating the management of depression in routine HIV care in Uganda

PMEN-D-23-00031R1

Dear Mr. Ssebunnya,

We are pleased to inform you that your manuscript 'Process evaluation of the HIV+D intervention for integrating the management of depression in routine HIV care in Uganda' has been provisionally accepted for publication in PLOS Mental Health.

Best regards,

Justus Uchenna Onu, MBBS, FWACP, FMCPsych

Academic Editor

PLOS Mental Health